# Medicinal Hypervalent Tellurium Prodrugs Bearing Different Ligands: A Comparative Study of the Chemical Profiles of AS101 and Its Halido Replaced Analogues

**DOI:** 10.3390/ijms23147505

**Published:** 2022-07-06

**Authors:** Lorenzo Chiaverini, Damiano Cirri, Iogann Tolbatov, Francesca Corsi, Ilaria Piano, Alessandro Marrone, Alessandro Pratesi, Tiziano Marzo, Diego La Mendola

**Affiliations:** 1Department of Pharmacy, University of Pisa, Via Bonanno Pisano, 6, 56126 Pisa, Italy; lorenzo.chiaverini@farm.unipi.it (L.C.); francesca.corsi@phd.unipi.it (F.C.); ilaria.piano@unipi.it (I.P.); diego.lamendola@unipi.it (D.L.M.); 2Department of Chemistry and Industrial Chemistry, University of Pisa, Via G. Moruzzi, 13, 56124 Pisa, Italy; damiano.cirri@dcci.unipi.it (D.C.); alessandro.pratesi@unipi.it (A.P.); 3Institute of Chemical Research of Catalonia (ICIQ), The Barcelona Institute of Science and Technology, 43007 Tarragona, Spain; 4Dipartimento di Farmacia, Università degli Studi “G. D’Annunzio” Chieti-Pescara, Via dei Vestini, 66100 Chieti, Italy; amarrone@unich.it

**Keywords:** tellurium, AS101, inorganic drugs, prodrugs, bioinorganic chemistry, coordination chemistry, clinical trials, metalloids

## Abstract

Ammonium trichloro (dioxoethylene-O,O′) tellurate (AS101) is a potent immunomodulator prodrug that, in recent years, entered various clinical trials and was tested for a variety of potential therapeutic applications. It has been demonstrated that AS101 quickly activates in aqueous milieu, producing TeOCl_3_^−^, which likely represents the pharmacologically active species. Here we report on the study of the activation process of AS101 and of two its analogues. After the synthesis and characterization of AS101 and its derivatives, we have carried out a comparative study through a combined experimental and computational analysis. Based on the obtained results, we describe here, for the first time, the detailed reaction that AS101 and its bromido- and iodido-replaced analogues undergo in presence of water, allowing the conversion of the original molecule to the likely true pharmacophore. Interestingly, moving down in the halogens’ group we observed a higher tendency to react, attributable to the ligands’ effect. The chemical and mechanistic implications of these meaningful differences are discussed.

## 1. Introduction

Inorganic chemistry has a key role in several fields of medicine and the use of inorganic molecules is of paramount importance for the diagnosis and treatment of several diseases [1,2]. Cisplatin and its analogues are extensively employed in cancer chemotherapy [3,4,5,6,7,8], while other compounds based on various transition metals entered clinical trials as promising antineoplastic agents [9,10,11,12,13,14,15]. Similarly, beyond cancer treatment, transition metals are exploited for applications in antimicrobial, antiparasitic, and antiviral therapy [16,17,18,19,20]. Alongside metals, metalloids—even known as semimetals—though only sporadically investigated, offer a rich and peculiar chemistry for their use in the preparation of compounds endowed with remarkable medicinal properties. Among metalloid-based drugs [21,22], hypervalent tellurium compounds (telluranes) are of utmost interest due to their promising therapeutic activity accompanied by a high tolerability [23,24]. Specifically, ammonium trichloro (dioxoethylene-O,O′) tellurate (AS101) is a potent immunomodulating agent that to date entered several clinical trials for treating psoriasis, genital warts, chemotherapy-induced thrombocytopenia, dermatitis, macular degeneration and also acute myeloid leukemia and HIV (see clinicaltrials.gov website) [21,25,26].

In this frame, AS101 represents the first Te-based agent reaching human experimentation [21,27]. Its beneficial biological effects mainly rely on the presence of the tellurium (IV) center capable of tightly interacting with thiol-bearing residues [24]. Accordingly, these interactions represent the molecular basis for the pharmacological activity of AS101 that potently and selectively inhibits the cysteine proteases in a time- and concentration-dependent manner [28]. The medicinal properties of AS101 spurred the interest for Te-based compounds and some examples of these compounds featured by various ligands appeared in the last years [23]. For instance, some authors reported on selected organotelluranes bearing different halide ligands. Interestingly, the same authors described that the analogues with bromide ligands were more reactive towards cysteine cathepsins V and S, compared with the chloride-bearing counterparts [29]. Based on these arguments, we were interested to investigate whether selective modifications -such as the replacement of the chloride ligands of AS101 with different halides- may affect the activation profile. Accordingly, we prepared the AS101 analogues featured by the replacement of chlorides with Br and I (Figure 1).

The selective replacement of the ligands coordinating the metal center of metallodrugs is a reliable strategy to confer specific chemico-physical and pharmacological properties to a given molecule. Indeed, it has been widely exploited in recent years obtaining important hints and can be easily used for both metal- as well as metalloid-based drugs. In fact, on the one hand, the comparative analysis of a halide-bearing parent inorganic drug and its halido-replaced counterparts may provide relevant mechanistic information on the role of the halides in terms of desired pharmacological effects [30,31]. On the other hand, the modulation of certain parameters may lead to the preparation of pharmacologically-improved analogues endowed with properties that are sometimes even ameliorated with respect to the original compound [17,31,32]. For instance, through these modifications, it is possible to change and control some features including the conversion of compounds in the active species that, in turn, may impact the cellular uptake, the bioavailability and other relevant biological aspects [33]. Thanks to the combined experimental and theoretical approach, we describe here the reaction that AS101 and its analogues undergo in aqueous environment, and we validate the mechanistic hypothesis by computational calculations, also determining the associated thermodynamic parameters.

## 2. Results and Discussion

Starting from AS101 -synthesized following the reported procedure- [34] we succeeded in obtaining the bromide and iodide derivatives through a direct replacement of chlorides by the addition of a stoichiometric amount of KBr or KI (Materials and Methods section for details). The replacement of chloride with bromide or iodide resulted in less stable compounds endowed with a higher tendency to undergo activation, i.e., the tendency of the original species to degrade, in analogy with AS101 [35]. This feature was already evident during the synthetic process requiring the careful control of various parameters including reaction times and the use of freshly anhydrous solvents [36]. To shed light on the differences in the activation profiles, we decided to use a combined experimental and theoretical approach capable to confirm that moving down in the 17th group, thus replacing the Cl^−^ ligands with Br^−^ or I^−^, an enhanced tendency to degrade was obtained. It is worth remembering that AS101 is a prodrug that exerts the pharmacological effects via the production of oxide species that are rapidly generated after administration in the biological environment [23]. Thus, through ^1^H NMR experiments, we preliminarily and comparatively assessed the occurrence of degradation of AS101 and its halido replaced analogues in presence of an increasing amount of water (2, 10 and 20 equivalents), as well as at increasing time intervals (15 min, 90 min, and 24 h). Results confirmed the previous observations. Indeed, higher amounts of the bromide and iodide analogues undergo degradation under these conditions and the differences with AS101 become prominent already in presence of 10 equivalents of water. Interestingly, while the degradation of compounds was affected by the water amount, no differences were found increasing the incubation times. Indeed, no further degradation of the starting compounds was observed after 15 min (Figure 2). The results support that, in these conditions, the conversion to the final species is substantially immediate upon addition of water. Furthermore, in analogy with AS101, the two halido replaced derivatives generate the corresponding oxides (general formula NH_4_TeOX_3_, where X stands for Cl, Br or I). FT-IR spectra of the isolated hydrolysis products show the disappearance of the Te-O stretching vibrations around 500 cm^−1^ and 600 cm^−1^ [36] and the formation of a new absorption band at 800 cm^−1^ attributable to the Te=O stretching [37] (See Appendix A for details).

In order to verify the above findings, we eventually performed the mechanistic analysis of the conversion of AS101 and its derivatives via DFT calculations [38]. Indeed, DFT has often and successfully been applied to describe the hydrolytic activation of metallodrugs [39,40,41]. The reaction mechanism that we investigated is presented in Figure 3a and, in more detail, at the top of Figure 3b.

The same mechanism describes well the hydrolysis of AS101 and its bromide and iodide analogues. It consists of three stages: two attacks of the water molecules, leading to the cleavage of the glycol and formation of TeX_3_(OH)_2_ (X stands for the halide, i.e., Cl, Br, or I), and the subsequent release of the water, concomitant with the formation of the Te=O double bond, thus yielding the hydrolysis product TeX_3_O. The first water attack is initiated by the water molecule approaching the Te center from the less sterically hindered direction and allowing the hydrogen bond between the attacking water molecules and the oxygen. The subsequent transition state (TS) is characterized by the concerted approach of water’s oxygen to Te and the detachment of glycol’s oxygen, made possible by the concomitant water proton transfer to it (Figure 4).

This TS has the Gibbs free energy of 22.2, 21.4, and 18.7 kcal/mol in the case of Cl^−^, Br^−^, and I-based complexes, respectively. The connected product–adduct (PA) TeX_3_(OH)(OCH_2_CH_2_OH), with a still quite high energy (around 4 kcal/mol lower than TS), subsequently rearranges in a low-lying minimum (GFE in the range of 6.6–7.3 kcal/mol for various complexes) corresponding to the RA species of the forthcoming step. The second barrier is characterized by a similar TS structure, with the attacking water also involved in the concomitant transfer of a proton to the oxygen bound to Te. The computed GFE of TS is slightly lower than the first barrier, with values of 16.7, 14.6, and 13.9 kcal/mol for the Cl-, Br-, and I-based complexes, respectively. This TS leads to the complete detachment of glycol which stays, nevertheless, connected via hydrogen bonds to the Te-bound hydroxyls in the corresponding PA species. Finally, these geminal OH groups eliminate a water molecule, forming the double bond Te=O. The calculated activation barriers for the detachment of water were 14.5, 13.7, and 14.4 kcal/mol for the complexes with Cl, Br, and I, respectively. Thus, our data indicate the first water attack as the rate-determining step. The calculated reaction GFE energies are 13.0, 9.9, and 9.4 kcal/mol for the investigated complexes.

These computational data can be interpreted based on the basicity of the bound glycol oxygen and how it is affected by the halogen bound to Te. Indeed, the oxygen basicity increases by decreasing the electronegativity of the bound halide; hence, in the Cl < Br < I trend, the reaction barriers follow the opposite trend, i.e., the hydrolysis of an I-based complex should be favored mostly, whereas the reaction of a Cl-based compound should be less favored. Indeed, these considerations agree excellently with the trends that we have found computationally as well as with the experimental data.

Concerning the consistency of the calculated energy profiles, we note that in the experimental conditions employed in this paper as well as those previously reported for the hydrolysis of AS101 [35], the water is in excess. Thus, by assuming a second-order kinetics (v = k[reactant][H_2_O]), the water excess would increase the observed reaction rate proportionally, and the observed activation free energy would result to be down-shifted by -RTln([H_2_O]/[reactant]). By assuming 20-fold excess of water, yielding an activation barrier decrease of 1.7 kcal/mol, the calculated activation free energies for the hydrolysis of Cl, Br, and I-based complexes would be 20.5, 19.7, and 17.0 kcal/mol, respectively (Appendix A).

Finally, to preliminary assess whether the differences in the activation profiles and in the conversion to the corresponding species that are produced affected the pharmacological activity, we performed a cellular study on retinal pigment epithelia cells (ARPE19). The obtained results confirmed that all the tested compounds are cytotoxic in the nM range. However, moving from AS101 to its iodide-bearing analogue, the cytotoxicity increases and a lower value of LC_50_ (Lethal Concentration 50; i.e., the dose capable to kill 50% of cells) is found (see materials and methods and Appendix A for further details). In fact, AS101 has a LC_50_ value of 25.11 nM whereas the iodide analogue is 2-folds more cytotoxic (LC_50_ = 13.40 nM). The bromide derivative has a LC_50_ closer to that of AS101.

## 3. Materials and Methods

### 3.1. General Statement

All reagents and solvents were purchased from Sigma-Aldrich (St. Louis, MO, USA) and used without further purification. Celite (Celite^®^545) was also purchased from Sigma-Aldrich. The obtained products were stored at −20 °C. NMR spectra were recorded at 293 K on a Bruker Avance II 400 MHz; chemical shifts (expressed in parts per million, ppm) were referenced to solvent residual peaks.

IR spectra were using dried KBr tablets. The tablets were prepared through hydraulic press compression (15 tons load) of a KBr dispersion of the investigated sample. The dispersion was prepared through the classical “mortar and pestle” technique. The spectra were recorded in the 2000–400 cm^−1^ region with a Perkin Elmer Spectrum 100 FT-IR spectrometer. Each spectrum was mediated among four different scans after preliminary background acquisition.

### 3.2. Synthesis of Ammonium Trichloro(Dioxoethylene-O,O′)Tellurate (AS101)

AS101 was prepared as described in the literature [34]. TeCl_4_ (2.6 g; 10 mmol) was refluxed under magnetic stirring with dry ethylene glycol (1.55 g, 25 mmol) in dry CH_3_CN at 80 °C for 4 h. A white precipitate forms during the reaction. The solution was filtered and the solid obtained washed with CH_3_CN, dried under vacuum and collected. Yield: 2.1 g (65%). Elemental Analysis (CHN): Found: C, 7.84; H, 2.73; N, 4.41. C_2_H_8_Cl_3_NO_2_Te requires: C, 7.70; H, 2.58; N, 4.49. ^1^HNMR (400 MHz; DMSO-d_6_) δ: 7.22 (4H; NH_4_^+^); 4.38 (s; 4H; CH_2_, coordinated ethylene glycol); 3.56 (br; H_2_O); 3.38 (s, CH_2_, free ethylene glycol, due to partial degradation of complex owing to water in the organic solvent). ^13^C{^1^H}NMR (100 MHz; DMSO-d_6_) δ: 67.48 (CH_2_).

### 3.3. Synthesis of Ammonium Tribromo(Dioxoethylene-O,O′)Tellurate

AS101 (312 mg; 1 mmol), prepared as described, was dissolved in 5 mL of MeOH. KBr (357 mg; 3 mmol) was then added, and the solution was magnetically stirred for 4 h. After a few minutes of stirring, a colorless precipitate started to form (KCl) and the solution turned yellow. The solution was filtered using celite and the solvent evaporated under reduced pressure. The bright orange solid obtained was dried under vacuum and collected. Yield: 275 mg (62%) Elemental Analysis (CHN): Found: C, 5.15; H, 1.62; N, 2.63 C_2_H_8_Br_3_NO_2_Te requires: C, 5.39; H, 1.81; N, 3.14. ^1^HNMR (400 MHz; DMSO-d_6_) δ: 7.20 (4H; NH_4_^+^); 4.43 (s; 4H; CH_2_, coordinated ethylene glycol). 3.63 (br, H_2_O); 3.37 (s, CH_2_; free ethylene glycol, due to partial degradation of complex owing to water in the organic solvent); 3.15 (methanol; reaction solvent residual). ^13^C{^1^H}NMR (100 MHz; DMSO-d_6_) δ: 67.84 (CH_2_).

### 3.4. Synthesis of Ammonium Triiodo(Dioxoethylene-O,O′)Tellurate

The compound was prepared similarly to the bromide analogue of AS101. AS101 (108 mg; 0.35 mmol) was dissolved in 5 mL of MeOH, then KI (172 mg; 1.04 mmol) was added, and the solution was magnetically stirred for 4 h. After a few minutes of stirring a colorless precipitate started to form (KCl) and the solution turned dark brown. The solution was filtered using celite and the solvent evaporated under reduced pressure. The brown solid obtained was dried under vacuum and collected. Yield: 162 mg (80%). Elemental Analysis (CHN): Found: C, 4.56; H, 1.72; N, 1.94 C_2_H_8_I_3_NO_2_Te requires: C, 4.10; H, 1.38; N, 2.39. ^1^HNMR (400 MHz; DMSO-d_6_) δ: 7.10 (4H; NH_4_^+^); 4.43 (s; 4H; CH_2_, coordinated ethylene glycol). 3.59 (br, H_2_O); 3.38 (s, CH_2_; free ethylene glycol, due to partial degradation of complex owing to water in the organic solvent); 3.16 (methanol; reaction solvent residual). ^13^C{^1^H}NMR (100 MHz; DMSO-d_6_) δ: 67.16 (CH_2_). The signal at 62.70 is attributable to the free ethylene glycol due to partial degradation of complex owing to water in the organic solvent.

### 3.5. NMR Activation Experiments

Activation studies on AS101 and its analogues were performed with a procedure similar to that reported by Albeck et al. [35].

A small quantity of powder (0.05 mmol) was dissolved in 400 µL of DMSO-d_6_ obtaining a 125 mM solution. The solution was then transferred into an NMR tube.

Three samples were prepared as described and an increasing amount of D_2_O was added in the NMR tubes: (1) 2 µL (2 eq.); (2) 10 µL (10 eq.); (3) 20 µL (20 eq.). ^1^HNMR spectra were then recorded for each sample at different times after the addition of D_2_O (15 min; 90 min; 24 h). The raw data were processed with Topspin 4.0.9 software. The FID was converted in frequency domain spectra with a line broadening of 0.3 Hz. The complex’s signal (4.37 ppm for AS101 and 4.43 ppm for both its bromide and iodide analogues) was integrated and calibrated to an arbitrary value of 100. At variance, the signal of released ethylene glycol (3.38 ppm) could not be integrated in a conventional way, due to the presence of a broad signal of H_2_O. For this reason, the integral value attributable to the free ethylene glycol was obtained as the difference between the right side and the left side integrals taken from the center of the water peak up to 0.5 ppm away. The residual concentration of the investigated complex was determined by solving a simple equation of the form x + kx = c, in which “k” was the ratio between the integral value of free ethylene glycol (obtained as described above) and the integral value of the unreacted complex (equal to 100 by default). “c” was the initial concentration of the investigated complex (125 mM). Three independent measurements were carried out for each point.

### 3.6. Computational Studies

Gaussian 16 [42] quantum chemistry package was employed for computations. Optimization of all structures in DMSO was performed by using the hybrid functional B3LYP [43,44] with the basis set D95V for the elements of the first row [45] and combination of Los Alamos effective core potential and the basis set DZ for Cl and Te [46]. Frequency computations verified the stationary nature of the minima and produced the zero-point energy (ZPE) and vibrational corrections to thermodynamic properties.

Intrinsic reaction coordinate (IRC) computations were utilized to determine the reagent-adduct (RA) and product-adduct (PA) minima linked to the transition states for each studied reaction step. Indeed, DFT allows accurate characterization of the structures and reaction pathways for the complexes with transition metals [47,48]. The density functional B3LYP is noted to yield good geometrical structures and to estimate precisely the electronic energies [9,39]. The SMD (Solvation Model based on Density) variation of the integral equation formalism (IEFPCM) was employed to account for solvation in DMSO [49]. This method is known to yield free energies with considerably smaller errors than continuum models, both for neutral and charged complexes, as recently demonstrated [50]. The activation enthalpies and free energies were calculated as the difference between TS and the lowest energy value amidst detached reactants (R) and RA for the attack of the first water molecule, and the difference between TS and RA for the second and the third barriers, while the reaction enthalpies and free energies were computed as the difference between R and detached products (P). Appendix A reports the activation barriers for the three-step hydrolysis of AS101 and its analogues. In each case, the first barrier was estimated with the R → TS free energy, being the energy of R always lower than RA (vide infra), whereas the RA → TS free energy values are used to estimate the second and the third activation barrier. Therefore, the molar excess of water with respect to the substrate was also considered as detailed in the Appendix A.

### 3.7. Cellular Studies

#### 3.7.1. Cell Culture

ARPE19 cells (ref. CRL-2302, ATCC Inc.) (Manassas, VA, USA) were cultured in Dulbecco′s Modified Eagle′s Medium/Nutrient Mixture F-12 Ham supplemented with 10% fetal bovine serum (FBS), 1% penicillin/streptomycin at 37 °C in a 95% O_2_, and 5% CO_2_ humidified atmosphere. The material used for cell cultures was purchased from Sigma-Aldrich (Merck, Darmstadt, Germany).

#### 3.7.2. Drug Stock Preparation

The compounds were synthesized as described above and stock solutions (10 mM) were prepared in DMSO and then diluted to working concentrations in DMEM.

#### 3.7.3. Cell Viability

Cell vitality was tested by using CellTiter 96 Aqueous—One solution Reagent (Promega, WI USA). The cells were seeded in a 96-well plate at a density of 1 × 10^4^ cells/well and incubated at 37 °C in 5% CO_2_. After 4 days, cells were treated with compounds at various concentration (1; 10; 20; 60; 120 nM). The next day, cells were treated with the One Solution Reagent and further incubated for 2 h at 37 °C, 5% CO_2_. A sigmoidal curve was graphed to show the dose-inhibition response, in which the % of cell viability represents the y-axis and the Log (Concentration) in the x-axis. Then, the median lethal dose (LC50) was determined from the curve using The Origin Lab 8.0 program (MicroCal, Northampton, MA, USA). Data are presented as the means ± SEMs. Statistical analyses were performed by using One-way ANOVA followed by Levene’s post-test.

## 4. Conclusions

Overall, we have investigated and reported, for the first time, the detailed activation mechanism of AS101 that leads to the production of the true pharmacophoric species TeOCl_3_^−^ [35]. Furthermore, we have provided the relevant thermodynamic parameters associated with the process. The complementary ^1^HNMR and theoretical approach we have used turned out to be useful to finely describe the chemical transformation that AS101 undergoes in presence of water. Additionally, we have studied how the replacement of chloride ligands of AS101 with different halides affects the conversion of the complexes to the corresponding oxides [35]. These findings are of relevance because they demonstrate how a small modification in the chemical structure of AS101 may imply differences in the activation profile and potentially different biological effects. Additionally, these studies may provide useful indications for the design of novel Te-based drugs. There are several examples demonstrating that the modulation of specific biochemical parameters through small structural modifications of established metallodrugs impact the pharmacological effects [33]. Accordingly, we have performed some preliminary cell experiments. The comparative evaluation of the activity of AS101 and its halido replaced analogues confirms that upon replacement of chloride a change in the biological activity occurs. Specifically, the LC_50_ for the iodide analogue is 2-folds lower compared with that of AS101 in ARPE19 cells (retinal pigment epithelia).

This result is important because AS101 entered several clinical trials for various applications including anticancer or ocular diseases therapy. Thus, we can speculate that the potent pharmacological effects of AS101 might be modulated and even improved through the suggested structural modifications. For instance, the effective doses exerting the desired pharmacological effects might be lower in the case of the iodide analogue. Based on these premises, we have now started additional biological comparative assessments on AS101 and its halido replaced analogues to deepen the investigation of the pharmacological profiles.

## Figures and Tables

**Figure 1 ijms-23-07505-f001:**
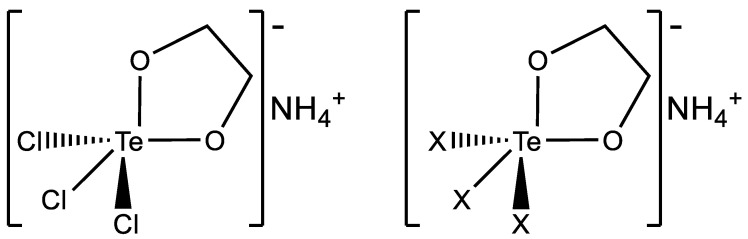
Chemical structure of AS101 (**left**) and the general structure of the analogues included in the present study (**right**). X stands for Br or I ligands.

**Figure 2 ijms-23-07505-f002:**
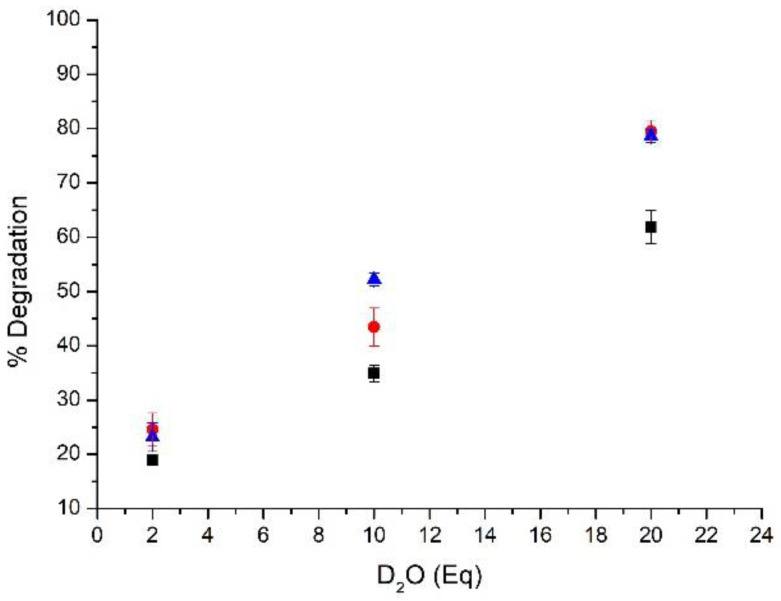
Results of the ^1^H NMR study of the degradation of AS101 (black) in comparison with its bromide and iodide analogues (red and blue respectively) in presence of increasing amounts of D_2_O. Solvent DMSO-d_6_, incubation time 15 min (results are reported as the average values of three independent measurements ± SD).

**Figure 3 ijms-23-07505-f003:**
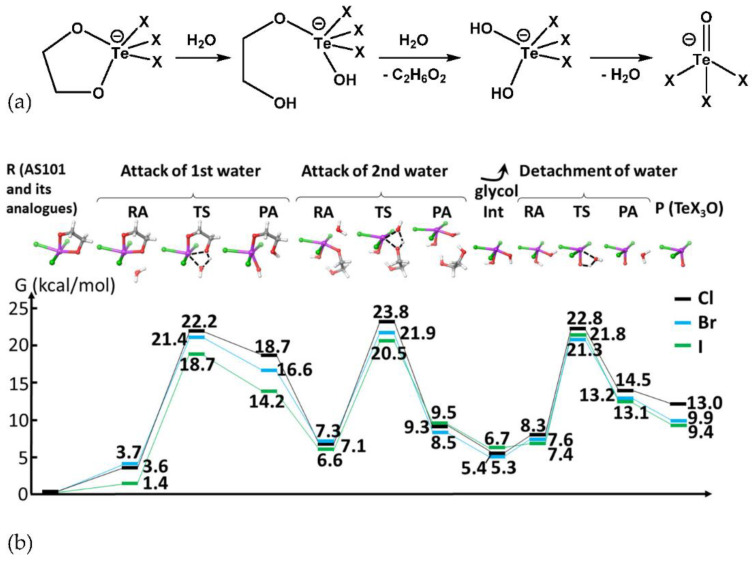
(**a**) general reaction mechanism for AS101 and its analogue in presence of water (X stands for halide ligands) and (**b**) computed reaction profiles for the hydrolysis of AS101 and its bromide and iodide derivatives. Gibbs free energy (GFE) values in kcal/mol are displayed. R, RA, TS, PA, Int, and P stand for reactant, reactant-adduct, transition state, product-adduct, intermediate, and product, respectively. Color scheme for the 3D structures above: Te (violet), Cl/Br/I (green), O (red), C (grey), and H (white).

**Figure 4 ijms-23-07505-f004:**
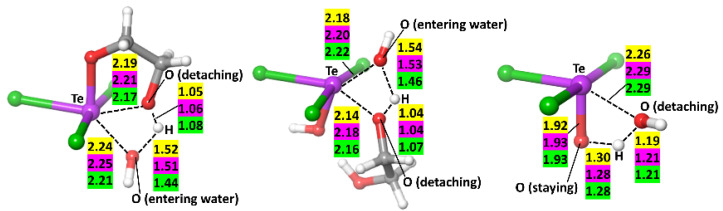
Transition states corresponding to the attack of the first and second water molecules (**left** and **center**, respectively) and the detachment of water (**right**). Distances between atoms are shown in angstroms. Distances corresponding to Cl, Br, and I are highlighted in yellow, pink, and green, respectively. Color scheme for atoms: Te (violet), Cl/Br/I (green), O (red), C (grey), and H (white).

## Data Availability

Not applicable.

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
