# Peer review of "Medicinal Hypervalent Tellurium Prodrugs Bearing Different Ligands: A Comparative Study of the Chemical Profiles of AS101 and Its Halido Replaced Analogues"

_ijms, 2022, doi:10.3390/ijms23147505_

Round 1
Reviewer 1 Report
Inorganic metal complexes play key role in several fields of medicine, and shows potential applications in antitumor, antimicrobial, antiparasitic, and antiviral therapy. In this manuscript, selective replacement of the chloride ligands of AS101 (ammonium trichloro (dioxoethylene-O,O') tellurate ), a potent immunomodulating agent, with different halides -Br and –I was studied and the possible mechanism of AS101 that leads to the production of the true pharmacophoric species TeOCl3− was analyzed with NMR spectra and DFT calculation. The studies are interested to the readers. However, significance of –Cl replacement with -Br and –I should be further clarified and additional experiments is needed.
1) The experimental data for the formation of double bond of Te-O is not enough. Is there any other method to prove it, for example characteristic IR or Laman vibration? The vibrational spectra could be speculated with DFT calculations.
2) What is the biological significance for the replacement of the chloride ligands of AS101 with –Br and –I. It is recommended that biological or cellular activity of Te complexes with different halides ligands could be tested and compared with each other.
3) “Indeed to the best of our knowledge- such studies have been never performed before.” in the Introduction can be moved.
Reviewer 2 Report
The authors report a comparative study of the chemical profiles of AS101 and its halido replaced analogues. The study is worth publishing in IJMS after addressing the points below:
1. Chemical structure of AS101 should be displayed in Fig.1 in addition to the general formula.
2. The quality of Figures 3 and 4 needs to be improved.
3. In the supplementary material file, integration should be added to all 1H NMR spectra
Round 2
Reviewer 1 Report
In the sixth line of 3.1, there are four spcae before IR spectra.
In the first line of 3.7.2, there is a space betwen 10 mM.